# Supersaturation and Precipitation Applicated in Drug Delivery Systems: Development Strategies and Evaluation Approaches

**DOI:** 10.3390/molecules28052212

**Published:** 2023-02-27

**Authors:** Yanxiong Gan, Jan P. A. Baak, Taijun Chen, Hua Ye, Wan Liao, Huixia Lv, Chuanbiao Wen, Shichao Zheng

**Affiliations:** 1School of Intelligent Medicine, Chengdu University of Traditional Chinese Medicine, Liutai Avenue 1166, Chengdu 611137, China; 2Dr. Med. Jan Baak AS, Risavegen 66, N-4056 Tananger, Norway; 3Department of Pathology, Section of Quantitative Molecular Pathology, Stavanger University Hospital, Gerd Ragna Bloch Thorsens Gate 8, N-4011 Stavanger, Norway; 4School of Pharmacy, Chengdu University of Traditional Chinese Medicine, Liutai Avenue 1166, Chengdu 611137, China; 5School of Pharmacy, China Pharmaceutical University, Longmian Avenue 639, Nanjing 211198, China

**Keywords:** supersaturation, precipitation inhibitors, dissolution, bioavailability, modeling

## Abstract

Supersaturation is a promising strategy to improve gastrointestinal absorption of poorly water-soluble drugs. Supersaturation is a metastable state and therefore dissolved drugs often quickly precipitate again. Precipitation inhibitors can prolong the metastable state. Supersaturating drug delivery systems (SDDS) are commonly formulated with precipitation inhibitors, hence the supersaturation is effectively prolonged for absorption, leading to improved bioavailability. This review summarizes the theory of and systemic insight into supersaturation, with the emphasis on biopharmaceutical aspects. Supersaturation research has developed from the generation of supersaturation (pH-shift, prodrug and SDDS) and the inhibition of precipitation (the mechanism of precipitation, the character of precipitation inhibitors and screening precipitation inhibitors). Then, the evaluation approaches to SDDS are discussed, including in vitro, in vivo and in silico studies and in vitro–in vivo correlations. In vitro aspects involve biorelevant medium, biomimetic apparatus and characterization instruments; in vivo aspects involve oral absorption, intestinal perfusion and intestinal content aspiration and in silico aspects involve molecular dynamics simulation and pharmacokinetic simulation. More physiological data of in vitro studies should be taken into account to simulate the in vivo environment. The supersaturation theory should be further completed, especially with regard to physiological conditions.

## 1. Introduction

Orally administered solid formulations have the advantage of good stability, low costs and manufacturing convenience. Solid formulations must be water-dissolved to pass the biological intestinal membrane. The drug concentration in the gastrointestinal (GI) tract is highly correlated to the absorption process. However, many new drugs are poorly soluble and have a very slow dissolution rate. The flux of a drug through the intestinal wall is directly proportional to the drug’s concentration and its permeability coefficient. Consequently, for the biopharmaceutical classification system (BCS) [1] class II drugs which have good permeability, poor solubility is the limiting absorption factor. It is increasingly challenging to obtain enough oral bioavailability in conventional dosage forms for these crystalline new molecular entities.

Drugs in solution may be at a concentration above their saturation solubility, that is, in a state of supersaturation [2]. This state is thermodynamically metastable, short-lived and the supersaturated drug can soon precipitate again. Precipitation inhibitors can delay the precipitation and may last long enough to supersaturate a certain drug in the intestine. This can result in increased and more rapid absorption of the drug molecule. The supersaturation and precipitation process can be explained by the classical “spring-parachute” model [3] (Figure 1). The figure points to a promising therapeutic strategy which, due to supersaturation, can improve oral bioavailability of poorly soluble drugs. To describe the extent of supersaturation, degree of supersaturation (DS) is defined as the ratio of temporary apparent concentration and thermodynamic equilibrium solubility. With dedicated techniques, proper DS and suitable precipitation inhibitor, supersaturating drug delivery systems (SDDS) could induce supersaturation of the drug and inhibit precipitation in the GI tract. This then will improve the bioavailability [4].

Many researchers have attempted to unveil the mechanism of supersaturation or apply supersaturation in a drug delivery system. In a meta-analysis of 61 articles on SDDS, the mean solubility, permeability and oral bioavailability were enhanced 26.7-, 3.1- and 5.59-fold, respectively, by SDDS formulations [5].

To better understand and control supersaturation for efficient development and evaluation of SDDS, this review describes:Generation of supersaturation;Inhibition of precipitation;Mechanism studies and evaluation approaches.

Emphasis is placed on biopharmaceutical aspects, providing perspectives with profundity and an easy-to-understand approach, from the generation of supersaturation to the inhibition of precipitation. Then, we conclude with proposed methods to simulate the in vivo environment and the areas where the supersaturation theory should be further developed.

## 2. Generation of Supersaturation

To increase the concentration relative to the equilibrium solubility of drug molecules, supersaturation can be generated by reduced solubility or fast dissolution. The solvent-shift method is one way to attain “reduced solubility induced supersaturation”. Many molecules have low solubility in water but high solubility in organic solvents such as Dimethyl Sulfoxide (DMSO). When the drug solution of DMSO is shifted into water, the solubility decreases for this solvent-shift processing, but the concentration is still high, above the current solubility in the mix solvent, so supersaturation is generated [6]. Though the solvent shift method is simple and comparable to pH shift method [7], solvent-shift can hardly take place in physiological conditions. On the contrary, GI tract naturally offers pH-shift condition for the pH sensitive molecules, especially for the alkaline components. In addition, water-soluble prodrugs can be rapidly converted by enzymes to hydrophobic drugs. Their aqueous thermodynamic solubility is low, but the converted drugs are maintained in an aqueous solution at supersaturated concentrations due to slow precipitation kinetics. However, prodrug studies have hardly taken notice or taken advantage of supersaturation. Here, we further discuss supersaturation induced by pH-shift and immediate release formulations.

### 2.1. pH-Shift

pH-shift is a special way for drugs with pH-dependent solubility to form supersaturation. Basic drugs dissolve well in acidic solution, but when pH of the solution shifts to basic, the solubility commonly decreases. Similarly, acidic drugs dissolve well in basic solution but precipitate in acidic solution. In normal physiological conditions, the pH increases tremendously from the stomach to the duodenum (see Table 1). For the basic drugs in the BCS II, this physiological condition provides a chance to dissolve a high concentration in acidic gastric juice. However, after the highly concentrated drug solution is transported to intestine, the solubility decreases, so the supersaturated status is formed. Here, precipitation inhibitors can be used to prolong the supersaturation (this action will be described in detail below).

The pH-shift experiments on drug supersaturation can be conducted by “dumping” or “pumping”. For the “dumping” method, the simulated gastric juice with drug was simply dumped into simulated intestinal juice. It means that pH-shift induced supersaturation was completed instantaneously. Ketoconazole, a poorly water-soluble weakly basic drug, was dissolved in pH 1.5 diluted phosphoric acid to simulate gastric conditions. When the pH value was changed to 6.5 by phosphate buffer, ketoconazole reached a DS of 170-fold [10]. In another case, Pazopanib was introduced into Simulated Gastric Fluid (SGF) for 20 min dissolution, subsequently transferred to the pH 6.5 phosphate buffer environments. The DS was determined to be approximately 600-fold and lasted 5 ± 2 min [11]. However, the stomach obviously does not empty in such a short time. It has long been known that gastric emptying conforms to a mathematic model of the first-order rate [12]. Namely, the empty rate is negatively correlated to the remaining stomach volume. In the “*pumping*” method, the simulated gastric juice with drug is gradually pumped in small portions into the intestinal juice [13]. This means that the pH-shifts and supersaturation is formed as a gradually changing process. There are many research studies on “pumping” methods (in Section 4.1.2 Biomimetic Apparatus).

### 2.2. Supersaturating Drug Delivery System (SDDS)

SDDS is an umbrella term that includes, for example, an amorphous solid dispersion (ASD) formulation and mesoporous-based system. It can be defined as systems that are able to present drugs to the GI tract at concentrations above their equilibrium solubility [2]. In addition, the increased concentrations must be maintained for sufficient time periods to allow for significantly increased absorption.

Three types of SDDS are widely reported: ASDs, self-emulsifying drug delivery system (SEDDS), and mesoporous-based systems. They have been compared in a meta-analysis [5], summarized in Table 2. Considering the easier preparation and steadier storage, based on our experience, ASD might be a balanced SDDS. Nevertheless, each kind of SDDS has special advantages.

#### 2.2.1. Amorphous Solid Dispersion (ASD)

ASD is one of the key formulation technologies that aid the development of poorly soluble drug candidates [14]. ASD often exhibits fast dissolution to form supersaturated solutions [15]. ASD commonly increases dissolution behavior and supersaturation of the drug once it is exposed to water. This is attributed to a number of factors, such as improved wettability of the drug by the polymer, minimal particle size of the drug, separation of individual drug particles by polymer particles, and subsequent prevention of drug precipitation upon contact with aqueous media.

Ashwathy et al. recently summarized the mechanism of phase separation in supersaturation [16]. Furthermore, liquid–liquid phase separation (LLPS) has been determined to be vital for supersaturation maintenance of ASD [17]. As the free drug concentration consistently rises, the original homogeneous system is transformed into a heterogeneous system composed of a supersaturated solution phase and a drug aggregation supercooled liquid phase. This process produces highly concentrated (nano) droplets/particles, which cannot be absorbed directly but may act as drug reservoirs shuttles, or also to carry drug molecules effectively in the mucus layer [18,19]. A study prepared paclitaxel/polymer amorphous solid dispersion by solvent casting to evaluate the effect of supersaturation on the oral absorption of paclitaxel. The dissolution rate was remarkably improved, reaching a DS of approximately 100 and maintaining it for more than 2 h, with a 1.78-fold increase in relative oral bioavailability [20]. Kyosuke et al. [18] studied the relevance of LLPS of supersaturated solution in oral absorption of albendazole from ASD. In the absence of the polymer, LLPS of Albendazole occurred at 1.4 µg/mL, which was more than 10-fold the solubility, whereas it increased to 7.2, 7.0, and 3.8 µg/mL with PVPVA, HPMCAS, and Eudragit, respectively. The LLPS concentration correlates well with AUC in rats (R^2^ = 0.9421). A better control over both the solid-state stability and supersaturation generation and maintenance will lead us to delivery systems having desirable and predictable properties and a practical option to solubilize the “difficult to solubilize” drugs [21].

However, ASDs often require a high weight percentage of carrier, often leading to high dosage volumes. To overcome the disadvantage and to improve drug load capacity, co-amorphous combination with two or more components is proposed. Co-amorphous formulations have emerged as a potential supersaturating drug delivery strategy for poorly soluble drugs [22,23,24]. Besides polymers, one component drug in the co-amorphous solid may also maintain the supersaturation of another component drug by intermolecular hydrogen bonds [25]. More physical stability and in vitro and in vivo performances of co-amorphous solids were reviewed by Qin et al. [26]. They have highlighted the role of molar ratio, molecular interaction, and mobility which affect the physical stability of co-amorphous solids.

#### 2.2.2. Self-Emulsifying Drug Delivery System (SEDDS)

SEDDS is a lipid-based formulation with supersaturation effect [27,28]. As a matter of course, it also inherited the solubilization effect from lipid-based formulations. With both effects, supersaturatable SEDDS is promising to improve the solubility of BCS II drugs. More than four SEDDS products are commercially available, and more supersaturatable SEDDS formulations have been explored for poorly water-soluble drugs (reported in publications). They provided a clear vision that the polymers used in these formulations prevent precipitation and emerge as a better option to improve the oral bioavailability and absorption [29]. Silymarin supersaturatable SEDDS selected Poloxamer 407 as the optimal precipitation inhibitor. The bioavailability improved to 7.6-fold compared with the branded product (Legalon^®^, Meda). The hydrogen bond between silybin and Poloxamer 407 might be one of the main interactions, leading to the enhanced DS [30]. Using precipitation inhibitor PVP, simvastatin SEDDS maintained a 2.5-fold higher DS after 15 min of digestion [31]. Meanwhile, digestion acts as a “trigger” in SEDDS for enhanced supersaturation. The solubilization/precipitation behavior is correlated with the DS. Precipitation inhibitors significantly increased supersaturation stabilization. At higher drug loads (80% saturation) and in lipid-free SEDDS, this effect was lost, suggesting that the ability to stabilize supersaturation in vitro may overestimate utility in vivo [32]. Heejun et al. [33] reviewed current status of supersaturatable SEDDS; they described the effects of various physiological factors and the potential interactions between PIs and lipid, lipase or lipid digested products on the in vivo performance of supersaturatable SEDDS.

#### 2.2.3. Mesoporous Material-Based Dosage Form

Mesoporous material-based dosage form refers to the material with a hole between 2~50 nm, with “finite-size effect”. That is, when the size of the pore is reduced to the molecular level, the characteristics of the solid mesoporous material will change greatly. With large surface area and tiny pore size, mesoporous material-based systems can dissolve quickly to generate supersaturation and significantly improve the maximum DS. Under sink conditions, the larger the pore is, the faster the drug release [34]. However, the pore size is suggested to be selected according to the molecular size for better stability [35].

The mesoporous material can not only be silica but also magnesium, calcium carbonate, etc. Silica is the most widely used mesoporous material. Ritonavir mesoporous silica delivery system led to a maximum DS of 7.8 within 5 min, which finally corresponded to an approximately 50% drug release, because mesoporous silica adsorbed additional ritonavir from the solution [36]. Another study reported that the interaction between ritonavir and the surface of mesoporous silica is too strong to achieve higher supersaturation [37]. Mesoporous magnesium carbonate is another new mesoporous material, used to improve dissolubility of the poorly soluble compounds celecoxib, cinnarizine and griseofulvin [38]. Supersaturation release profiles were observed, and the areas under the concentration-time curves were 25- (celecoxib), 5- (cinnarizine) and 2-fold (griseofulvin) greater than those of the crystalline drugs. Celecoxib mesoporous magnesium carbonate resulted in both a higher dissolution rate and supersaturation of the substance in vitro as well as an increased transfer of celecoxib over a Caco-2 cell membrane [39]. A novel mesoporous calcium carbonate has a pore volume of 0.86 cm^3^/g, the pore-size distribution is centered at 7.3 nm and the surface area is over 600 m^2^/g. Model drug celecoxib enhanced release rates up to 65-fold more than the crystalline form, forming supersaturation and maintaining it for 2 h [40].

## 3. Inhibition of Precipitation

As of yet, precipitation inhibitors have not been clearly defined. Researchers generally call the excipient which could inhibit precipitation “precipitation inhibitors”, regardless of whether the excipient prolongs supersaturation status by inhibiting drug molecular aggregation to reduce precipitation, or by improving apparent solubility to decrease DS. To inspire deeper cognition on precipitation inhibitors, this chapter refers to the mechanism of precipitation inhibition, reviewing the character of several commonly used precipitation inhibitors (such as Hydroxypropyl methylcellulose (HPMC), Hydroxypropylmethylcellulose-acetate succinate (HPMCAS), Polyvinyl pyrrolidone (PVP) and Cyclodextrins, as shown in Figure 2) and finally discusses screening of precipitation inhibitors.

### 3.1. Mechanism of Precipitation Inhibition

Recent studies have preliminarily revealed some theories and mechanisms of supersaturation. As Warren et al. [41] and Price et al. [42] summarized, the basic mechanism of supersaturation regards the following aspects:(1)Enhancing viscosity to reduce molecular motion (hindering nucleation) and diffusion coefficient (impeding crystal growth);.(2)Improving mass-liquid surface energy;(3)Changing the absorption layer of the crystal surface, such as adhering to the crystal surface to block the growth of the crystal and influence the crystal properties;(4)Controlling the dissolution of the crystal surface, thus influencing the molecules to concentrate in the crystal;(5)Increasing solubility to decrease DS.

In addition, there are some further mechanisms reported. For example, precipitation inhibitors induce the precipitation to amorphous, hence the precipitation re-dissolves faster than the common crystal form [43]; hydrophilic polymers could not only inhibit crystal growth in supersaturated solutions, but also prevent aggregation [44]; macrocycles, which include crown ethers such as cyclodextrins, can host molecules forming supramolecular complexes, the host–guest interaction inhibit the aggregation of the drug molecules [45]. These details sometimes contribute significantly to supersaturation.

### 3.2. Character of Precipitation Inhibitors

#### 3.2.1. Hydroxypropyl Methylcellulose (HPMC)

HPMC is the propylene glycol ether of methyl cellulose, hydroxypropyl and methyl combined with an anhydrous glucose ring by ether bond. As the degree of polymerization is variable, a series of HPMCs were named with different molecular weight. This character endows different viscosities of HPMC, leading to different properties of supersaturation and precipitation, because hydrogen bonding is one of the main forces for HPMC to inhibit precipitation [46]. For example, in a study comparing the ability of precipitation inhibition of three HPMCs, the result showed that nucleation inhibition K4M > K100M > K15M, and crystal growth inhibition K100M > K4M ≈ K15M [47]. Even at low concentrations (0.5 μg/mL), HPMC is able to retard felodipine nucleation and keep its solution supersaturated for approximately 100 min. The supersaturation period could last up to 1000 min in the presence of 3.5 μg/mL of HPMC but reduces to only 15 min in the absence of polymer [48]. Another interesting study [49] reported HPMC significantly increased the dissolution and notably prolonged supersaturation of naringenin and isonicotinamide co-crystals but had limited effect on the naringenin crude drug. HPMC further elevated C_max_ and AUC by 2.2 times and 3.7 times. HPMC was selected as a precipitation inhibitor in a self-microemulsifying drug delivery system, for its improvement to the dissolution and cellular uptake in astaxanthin [50].

#### 3.2.2. Hydroxypropylmethylcellulose-Acetate Succinate (HPMCAS)

HPMCAS is one of the best precipitation inhibitors. It has been reported that HPMCAS is the most effective inhibitor in maintaining drug supersaturation among 41 polymeric materials including polymers and surfactants, carboxylic acid, etc. [51,52]. HPMCAS suppressed crystallization of nifedipine, carbamazepine, mefenamic acid and dexamethasone [53]. HPMCAS with a higher succinoyl substituent ratio increased the dissolution rate and strongly suppressed the drug crystallization of nifedipine. The inhibition of crystallization was affected by pH, with the carbamazepine crystallization being inhibited at a higher pH due to the hydrophilization of HPMCAS derived from succinoyl ionization. Recently, Masafumi et al. [54] confirmed that HPMCAS is able to inhibit liquid–liquid phase separation in the supersaturation process of naftopidil amorphous solid dispersion at 41 µg/mL. Sheshank and Vikas [55] evaluated HPMCAS in atazanavir in silico; they discovered that their complex possesses low binding energy (−2.98 kcal/mol) and attains the stability at the earliest with a root-mean-square deviation (RMSD) fluctuations less than 1.5 Å, indicating that HPMCAS had strong inhibitory effect on drug precipitation.

However, HPMCAS is an enteric polymer, dissolved in high pH intestinal fluid but not in low pH gastric juice, which limited in vivo absorption [56]. In other words, HPMCAS is not suitable as an active pharmaceutical ingredient (API) to be absorbed in the stomach or upper intestine (especially for weak basic drugs). Enteric soluble polymers Hydroxypropyl methylcellulose phthalate, cellulose acetate phthalate and Eudragit^®^ have the same disadvantages.

#### 3.2.3. Polyvinyl Pyrrolidone (PVP)

PVP is a commonly used binder in many pharmaceutical tablets. Owing to its polarity, PVP binds exceptionally well to polar molecules. This makes PVP an excellent precipitation inhibitor. Through molecular docking and ^1^H NMR analysis, Zhang et al. [57] proved that PVP provides H-acceptors for hydrogen bonds to naringenin. The strong intermolecular interaction between PVP and naringenin can prevent the self-aggregation of the drug molecules, delay the formation of crystal nuclei, and inhibit the crystallization. Drug–polymer combinations capable of hydrogen-bonding in the soluble state (dipyridamole-PVP) are more effective in preventing drug crystallization compared to the drug–polymer systems without such interaction (Cinnarizine-PVP) by implementing binary and the ternary Flory–Huggins theory [58]. In addition, PVP is applied as carrier in ASD to generate supersaturation. In the presence of PVP, amorphous zafirlukast maintained a supersaturated state lasting more than 20 h in a fasting state intestinal medium. Moreover, the concentration of drug dissolved in the supersaturated state increased [59]. PVP can also be used in cocrystal to improve the dissolution rate and enhance solubility. As Eunmi et al. reported [60], the solubility of emodin–nicotinamide cocrystal was 2.07-fold higher than that of emodin, and the percent of emodin dissolved at 15 min was increased 2.88% by PVP. PVP was reformed to variant like polyvinylpyrrolidone—vinyl acetate copolymer (PVP/VA) and polyvinyl acetate (PVAc). The two variants increased supersaturation of celecoxib from 2.0-fold to 2.4- and 10.0-fold [61].

#### 3.2.4. Cyclodextrins

Cyclodextrins are hydrophobic inside and hydrophilic outside, hence can form complexes with hydrophobic compounds to increase the concentration. As supramolecular chemistry developed in recent years, the complexes of cyclodextrins and drug are recognized as host–guest molecular structures [62,63]. Ayako et al. [64] determined that hydroxypropyl-β-cyclodextrin surrounds indomethacin, enhances its dissolution rate and increases its equilibrium concentration, causing supersaturation (spring) and its sustaining deployment (parachute). Besides inclusion complexes, cyclodextrins also form non-inclusion complexes, as shown in Figure 3. The hydroxy drugs on the outer surface of the cyclodextrin molecule are able to form hydrogen bonds with other molecules, and cyclodextrins can, like non-cyclic oligosaccharides and polysaccharides, form water-soluble complexes with lipophilic water-insoluble compounds, that is, non-inclusion complexes [65]. However, the increased concentrations by cyclodextrins may not enhance absorption, since inclusion in cyclodextrins may limit the drug available to permeate through the intestinal epithelium [66]. Therefore, it is important to consider the combined strength between cyclodextrins and drugs to avoid the over-strong supramolecular complexes.

#### 3.2.5. Others Precipitation Inhibitors

Soluplus^®^, a polyvinyl caprolactam–polyvinyl acetate–polyethylene glycol-based graft copolymer, is also a nonionic surfactant. A Soluplus^®^-based S-SEDDS retarded tacrolimus precipitation and maintained >80% of the accumulated dissolution rate for 24 h [67]. The micelles formed by Soluplus^®^ enwrapped the molecular itraconazole inside the core, which promoted the amount of free drug in the intestinal cavity and carried itraconazole through the aqueous boundary layer, resulting in high absorption by passive transportation across biological membranes [68].

Eudragit^®^ polymer systems, with poly (meth) acrylate chemistry, provide formulators with an exceptionally versatile platform for designing drug delivery to match the specifics of individual pharmaceutical activities and treatments. For example, Eudragit^®^ formed glass solutions and prolonged supersaturation of weak acid model compounds indomethacin and naproxen. The drug–polymer combination precipitation remained amorphous for 24 h [69].

Besides the polymers, the components in the co-amorphous may also affect the supersaturation. In the dissolution test of co-amorphous ritonavir-saccharin, saccharin prolonged the supersaturation of ritonavir from 90 min to 300 min, because the ritonavir thiazole group was bonded with the saccharin amine group [70].

As the emerging synthon customized excipients, there will be a trend for drug developers to design a special precipitation inhibitor for a unique poorly soluble API to formulate a supersaturation drug delivery system [71].

### 3.3. Screening Precipitation Inhibitors

Diverse precipitation inhibitors exhibit different affinity to each API. The excipient-induced supersaturation was illuminated in a solvent quench assay for 39 compounds. The results showed that the supersaturation degree can vary enormously, from 1 to 10,000 [72,73]. For example, the effects of molecular weight were prominently seen in the case of HPMC but not in PVP. This may be due to interaction between a drug and a polymer [74]. Another report showed that only modest excipient-mediated stabilizing effects on supersaturation were observed using HPMC-E5 and Eudragit^®^, whereas PVP K25 exerted no effect [75]. Drug developers commonly screen a suitable precipitation inhibitor for a drug by trial and error.

A high-throughput screening method was adopted to select precipitation inhibitors by the 96-well plate-based microplate reader technique [76]. It is indeed more efficient when researchers can quickly and judiciously pick out the best excipient. A novel screening protocol was developed to select precipitation inhibitors for supersaturating formulations by calculating drug–polymer mixing enthalpy [77]. The mixing enthalpy was calculated by Conductor-like Screening Model for Real Solvents (COSMO-RS). The results showed a strong positive correlation between the drug–polymer mixing enthalpy and the overall formulation performance.

## 4. Mechanism Studies and Evaluation Approaches

Better understanding and control of supersaturation is desirable for better in vivo performance and better bioavailability. At an early stage, researchers focused on the basic phenomenon of supersaturation, nearly solving the problems in vitro, such as different behavior in different pH [78] and the drug–polymer interaction [79]. However, the real intraluminal condition where the supersaturation occurs in vivo is quite different, even between fasting and feeding state. On the one hand, the GI tract secretes different digestive juices at different anatomical parts. Consequently, the GI fluid varies with the locally prevailing pH. The variation of pH affects hydrogen bond, viscosity and surface tension. On the other hand, the hydrodynamics (especially in the stomach) and permeability (especially in the intestine) make in vivo reality much more complex. Subsequent research studies attempted to simulate the real in vivo situation and mediate the concision to meet the biomimetic environment. This section illustrates the supersaturation biomimetic mechanisms operating in specific research aspects in vitro, in vivo and in silico.

### 4.1. In Vitro Studies

#### 4.1.1. Biorelevant Media

Biorelevant media are an important part of biomimetic environment for supersaturation study. They consist of simulated physiological compositions and therefore drug product-independent media. Determining solubility and supersaturation in media is very relevant to the physiological environment, as solubility determines absorption. The choice of biorelevant media may influence the behavior of supersaturating formulations [80]. The development of biorelevant media has undergone different important steps, from phosphate and hydrochloride buffers to the five common bile salts. The content and properties of different biorelevant media are shown in Table 3.

Previously, SIF (Simulated Intestinal Fluid) and SGF (Simulated Gastric Fluid) were composed of phosphate and hydrochloride buffers according to U.S. Pharmacopeia (USP), but usually without pepsin and pancreatin for nonprotein drugs. Now, similar phosphate buffer is still used in supersaturation studies [85].

Then, FaSSIF/FeSSIF/FaSSGF (Fasted/Fed State Simulated Intestinal/Gastric Fluid) were composed of appropriate sodium taurocholate and lecithin based on SIF and SGF, according to data from Dressman’s research group [81]. The supersaturation of different drugs in fed or fasted simulated intestinal fluid is quite different. For example, fed state promotes itraconazole but depresses etravirine and loviride [86]. Therefore, both fed and fasted state biorelevant media should be mentioned in research studies.

Recently, Fe/FaSSIF-V2 and Fe/FaSSCoF were improved to mimic the human intestinal juice more closely. Within 0.2 mM lecithin, FaSSIF-V2 has been demonstrated to closely represent Human Intestinal Fluid (HIF) [87]. The main improvement is the reduced amount of lecithin and recruitment of maleic acid for buffer, leading to lower osmolality. Hence, FaSSIF-V2 provides lower drug solubility except for acidic drugs [88].

A new version, FaSSIF-V3, is coming. Considering the variation of the bile salt in the cholate, Dressman’s group prepared FaSSIF-V3 with each of the five common bile salts (as shown in Table 3) [83]. Yet a study compared the dissolution behavior of various drugs in different FaSSIF versions, and concluded the differences are minor [89].

In addition, pepsin can maintain the supersaturation solubility of a poorly soluble weak base by nucleation inhibition [90]. In addition, intestinal mucus was also reported to be capable of stabilizing supersaturation of poorly-water soluble drugs [91]. Analogously, the question can be raised whether trypsin or other GI secretion enzymes may play a role in drug supersaturation. It is interesting to further explore and complete this field of research.

In fact, intestinal fluid is mainly buffered by hydrogen carbonate ions [92]. Therefore, the bicarbonate buffer was used to improve prediction of in vivo supersaturation and precipitation. Compared with phosphate-buffered FaSSIF, bicarbonate-buffered FaSSIF exhibited a better predictive power [93]. In addition, Hank’s buffer and Krebs buffer were developed to mimic the ionic strength and buffer capacity of intestinal fluid. However, bicarbonate buffers are restricted by lack of practicability and poor reproducibility of the results [94], and the inherent difficulties in volatility associated with CO_2_ make bicarbonate buffers difficult for routine dissolution testing [95].

Solubility assessment in HIF could be considered as the “gold standard”, although HIF is not easy to obtain. Augustijns’ research group has conducted many supersaturation studies on HIF. They first aspirated HIF from healthy volunteers to simulate the supersaturation behavior of fosamprenavir [96]. Others also aspirated HIF [97] from the duodenum and HGF [75] from the stomach in fasted and fed states to evaluate the supersaturation in human digestive juice.

#### 4.1.2. Biomimetic Apparatus

Biomimetic apparatus is another important part of supersaturation environment research. USP dissolution apparatuses are the most commonly used methods for drug dissolution, but they are neither good enough at mimicking transportation nor absorption. Researchers have developed a variety of in vitro apparatus to simulate the physiological environments for supersaturating process investigation. Pharmaceutical Education and Research with Regulatory Links (PEARRL) project even emphasized oral biopharmaceutics tools [98].

In the early stage, the instruments of in vitro approaches used to simulate GI transfer and assess epithelial permeability of drugs from pharmaceutical formulations, which were summarized [99] (Figure 4). More recently, USP II dissolution apparatus and USP IV apparatus were combined, with organic phase consisting of octanol as absorption apartment [100]. The result exhibited a complex interplay among dissolution, precipitation, and partition. Though the biphasic dissolution so widely used until nowadays is easy to conduct [101], a shortcoming is that the precipitation may enter the octanol layer, leading to over estimation of absorption. A filter membrane could be amended between the two phases to avoid the mix [102].

Varieties of permeable membranes have been reported. With a regenerated cellulose membrane, an artificial membrane insert system predicted the effects of pH and food intake on absorption [103]. Hydrophobic treated polyethersulfone was adopted as biomimetic membrane, whose AUC was not significantly different from that of porcine intestinal tissue [104]. Parallel artificial membrane permeability assay (PAMPA) is a relatively mature approach to test the permeability of a drug. By using an artificial membrane separating side-by-side chambers, a study selected optimal permeability properties according to the strategy of high supersaturation gradient promoting permeation [105]. Another study observed supersaturation increased mass flow rate through PAMPA [106]. Permeapad^®^ was demonstrated with apparent permeability coefficient to the absolute bioavailability of metoprolol administered buccally to mini-pigs [107]. In addition, a Caco-2 based Transwell was also applied to determine that the impact of HPMC on loviride transport was inferior to its precipitation inhibitory capacity in a non-absorption environment [108].

Many apparatuses attempt to simulate gastrointestinal transportation. A simulated stomach duodenum (SSD) model with flow rate simulated the fasted upper GI tract, as shown in Figure 5 [6]. SSD model mimicked the physiological and dynamic environment of the GI tract well. The basal gastric volume was 50 mL and duodenal volume was 30 mL. The simulative models were used to elucidate and explore the significance of the effect of different physiological parameters and additives on drug supersaturation. In addition, Stomach-to-Intestine Fluid Changing System simulated the pH shift in the gastrointestinal tract, by which the variable plasma exposure was explained according to the in vitro sensitivity analysis [109].

The Gastrointestinal Simulator (GIS) [110,111,112] consists of three dissolution chambers representing the stomach, the duodenum, and the jejunum. Based on GIS, an in vitro–in silico–in vivo approach was developed to screen SDDS by ranking these formulations toward their in vivo performance [113]. GIS has also been used to study the precipitation pathways of dexketoprofen trometamol salt, and the data from GIS in vitro correlated well with in vivo data [114]. Based on GIS, we developed Biphasic Gastrointestinal Simulator (BGIS), as shown in Figure 6, to improve the evaluation of supersaturation in vitro [115]. Octanol was added to both the duodenal chamber and the jejunum chamber as absorption phases. In BGIS, the supersaturation degree of apatinib was 4.99 and 1.12 in the duodenal chamber and the jejunum chamber, respectively. The mean prediction errors of pharmacokinetics compared with the in vivo data were 4.58% (C_max_) and 1.93% (AUC).

The next generation of gastrointestinal simulator could be more biorelevant by promoting “pulsing”, because the stomach empties the content not only in a first order rate but also by pulse [116]. A pulsed-packet gastric fluid emptying model has also been used, defined as a function of gastric motility using a Poisson point process with motility-dependent intensity. This model accounts well for the average observed emptying rates and encompasses the variability of observed volume and dosage form emptying rates [117]. Furthermore, gastrointestinal organoids based on microfluidic chips are promising to be designed and applied to study drug absorption and supersaturation.

### 4.2. In Vivo Studies

It is not easy to assess supersaturation in vivo directly, especially in human beings, unless aspirating the supersaturating drug in the gastrointestinal tract. Most researchers evaluate supersaturation in biological bodies by oral administration then measuring the drug blood concentration. Some supersaturation studies conducted on animals applied intestinal perfusion.

#### 4.2.1. Oral Absorption

The commonly used pharmacokinetic study animals are rats and dogs. Rabbits [118] and pigs [119] have also been used in supersaturatable formulation bioavailability studies. Rodents are easy to breed but not easy to administrate solid supersaturatable formulation to. Supersaturatable SMEDDS could be mixed homogeneously with water before gavage. For ASD and mesoporous material-based dosage forms, it is better to choose dogs or pigs for oral administration.

For example, Sprague–Dawley rats [120,121] and Wistar rats [122] were routinely administrated pre-emulsified supersaturatable SMEDDS formulations by oral gavage, then blood samples were collected at designated times. Itraconazole supersaturatable ASD was evaluated in beagle dogs. The dogs were administered orally with hard capsules then blood samples were collected in 24 h. Compared with a commercial product Sporanox^®^, the supersaturatable ASD increased 1.8-fold in the AUC [123].

#### 4.2.2. Intestinal Perfusion

Intestinal perfusion could not only offer access to permeability and absorption data as commonly used, but also provide an approach to measure the drug supersaturation in the gastrointestinal tract. The perfusate should adopt the bio-relevant media to simulate the real condition in vivo. In addition, the intestinal perfusion study on supersaturation should be conducted in a single pass, but not in circulation, because the effluent would dilute and interrupt the supersaturation.

Different parts of the gastrointestinal tract could be perfused, respectively. For example, Sprague–Dawley rats were used to conduct in situ single pass jejunal perfusion experiments. Simultaneously, mesenteric blood was collected to assess drug flux across the corresponding isolated segment of the rats’ jejunum, without the confounding effects of hepatic first-pass metabolism. Increased fenofibrate supersaturation resulted in increased drug exposure [124].

The intestinal infusion can be combined with an in vitro dissolution apparatus to simulate the upper digestive tract. For example, C57BL/c mice were perfused from the proximal jejunum by the fluid in the duodenum chamber of the GIS. This study indicated that DS observed in the GIS enhanced dipyridamole and ketoconazole absorption [111].

#### 4.2.3. Intestinal Content Aspiration

Aspirating the intestinal content with dissolved drug for concentration determination is the most direct way to access the supersaturation and precipitation process in the human gastrointestinal tract. Professor Augustijns’ research group is a main force in this field. They introduced double-lumen polyvinyl catheters via the mouth/nose and positioned in the duodenum of the small intestine and in the antrum of the stomach. Fluoroscopic imaging was used to confirm the position of the catheters. Formulations were intragastrically administered, and then gastric and duodenal fluids were aspirated for determination of dissolved and total posaconazole [125]. In a similar approach, they investigated the impact of relevant GI conditions (fasted state, fed state and fasted state with concomitant proton pump inhibitor) on the intraluminal dissolution and supersaturation behavior of indinavir [126].

We propose that fiber optic detection system can be introduced into the intestinal content in situ detection such as endoscope. Of course, fiber optical probe should be redesigned as flexible and long enough to fit the access into gastrointestinal tract.

### 4.3. In Silico Studies

With the development of computer science, a lot of software is emerging to conduct in silico studies. For supersaturation and precipitation inhibition mechanism studies, software programs GROMACS, Gaussian and Materials Studio are used for molecular modeling, and GastroPlus^™^, Simcyp^®^ and GI-Sim are used for pharmacokinetics simulations. With the support of such professional software programs, it is possible to analyze the intermolecular interactions and also forecast the effect of polymer on the API, thus enabling researchers to accurately select the optimized excipients for SDDS. Additionally, mathematical models can also help predict supersaturation and in vivo absorption [127].

#### 4.3.1. Molecular Modeling

Molecular dynamics simulation studies between drugs and different polymers are valuable to understand the mechanisms of supersaturation on molecular level. The commonly used Software includes GROMACS, Gaussian and Materials Studio. A former study employed GROMACS to simulate the molecular dynamics between drugs and HPMC polymers. The snapshots are shown in Figure 7. It was pointed out that HPMC stabilizes indapamide better than glibenclamide through the higher number of hydrogen bonds formed [128]. Another study used GROMACS to analyze molecular dynamics between drug and bile salts. When the drug interacts with the polar faces of the bile salt, nevirapine stayed in closer contact with sodium hyodeoxycholate than with sodium cholate. Nevirapine−hyodeoxycholate tended to form tightly arranged complexes, which may explain the predisposition of hyodeoxycholate to form intermolecular hydrogen bonds during the trajectory. From 100 ns molecular dynamics trajectories, hyodeoxycholate interacts with nevirapine for the majority of the trajectory, through van der Waals and hydrogen-bonded interactions [129]. Gaussian 09 software, based on quantum mechanical molecular model, calculated the hydrogen bond between the polymer and the drug accompanied by 5 to 8 kcal/mol increase in binding energy. The tertiary amines and the carboxylate groups via a proton have the strongest binding energy ranging from 20.0 to 27.9 kcal/mol [130,131].

Recently, Materials Studio was used to determine the theoretical interaction strength among emodin, nicotinamide and Kollidon^®^ VA 64, and to elucidate potential mechanisms of the supersaturation [132]. The snapshots are shown in Figure 8. The diffusion coefficient (representing the molecular lateral movement of drug) of Emodin-Kollidon^®^ VA 64 system and Emodin-Kollidon^®^ VA 64-Nicotinamide system are 1.63 × 10^−1^ and 1.48 × 10^−1^ m^2^/s, respectively. The cohesive energy density (representing the interaction strength of drug) values of the tow systems are 8.06 × 10^7^ and 6.95 × 10^8^ J/m^3^, respectively. This low molecular mobility and strong intermolecular interaction prevented spontaneous drug aggregation, thus maintaining the supersaturation of the system. The result was verified by a serious experiment.

#### 4.3.2. Pharmacokinetic Simulation

GastroPlus^™^ is developed based on advanced compartmental absorption and transit (ACAT) model, which is composed of nine chambers of gastrointestinal tract, considering six states of drug in the dissolution and absorption. With this advantage on the meticulous physiological condition, it provides single or multiple first-order exponential precipitation or a complete mechanistic nucleation and growth model that can account for formulation effects due to nucleation inhibitors and solubilizers [133]. When using the mechanistic nucleation and growth model, the simulation outputs include the size, time and DS for initial particle formation, as well as the maximum DS achieved. For example, researchers applied a dynamic fluid and pH model in GastroPlus^™^ to simulate intraluminal and systemic concentrations of posaconazole in a biorelevant manner. The result indicated the simulated maximum DS was slightly overestimated [134]. However, there are some disadvantages: the study ignored some special forms of drugs, such as the salt and stereoisomers; many drug parameters are required to establish the model; the accuracy of the model is highly dependent on the modeling strategies [135].

Simcyp^®^ is based on advanced dissolution absorption and metabolism (ADAM) model, which is composed of seven chambers of gastrointestinal tract. Its advantage consists in considering absorption kinetics with respect to disintegration time, DS and precipitation rate [136]. The outcome was contrasted with that based on the more usual modeling assumption of a simple first-order absorption input and the use of average kinetic parameters. Its disadvantages were reported as the following: default models and parameters were allowed, potentially leading to wrong conclusions; it was unable to distinguish between the drug salt and the base present in different proportion, leading to more probable failure in BE testing [137].

GI-Sim is also based on ADAM model. Its advantage includes consideration of particle growth, amorphous or crystalline precipitation and supersaturation parameter. However, it was usually used with the above software as comparison. A study compared them, concluding that GI-Sim 4.1 and GastroPlus 8.0 performed better than Simcyp 13.1 in predicting the intestinal absorption of the incompletely absorbed drugs [138]. Sceptics commented that it had a number of pitfalls. For example, it accepted default values of supersaturation- and precipitation-related parameters without distinction [139]. The authors replied that the variation of supersaturation and precipitation behavior is indeed so wide that even an experimental data set cannot be successfully used [140]. Therefore, its disadvantage is also similar to that of Simcyp^®^.

Other publications on the application of physiologically-based pharmacokinetic (PBPK) models predict absorption of supersaturation drugs. Stella software was also used to predict PBPK. The PBPK modelling indicated that intragastric processes had significant impact on precipitation kinetics [141]. Furthermore, it is well acknowledged that incorporation of processes such as supersaturation/precipitation and presence of multiple API phases is not readily achievable with currently available PBPK models [142].

### 4.4. In Vitro–In Vivo Correlation (IVIVC)

In vitro–in vivo correlation (IVIVC) can be traced back to the FDA’s “Critical Path Opportunities for Generic Drugs”. It recognizes that designing better absorption models and developing of IVIVC are critical modeling and simulation research areas [143]. Predictive models of drug release profiles and the relationship between dissolution and bioavailability/bioequivalence can help to guide drug applicants in the implementation of quality by design, including SDDS [144].

Ordinary dissolution tests are performed under sink conditions for establishing IVIVC. However, supersaturation dissolution tests are obviously under non-sink conditions. IVIVC usually follows after in vitro and in vivo studies, but only a few studies successfully established level A IVIVC. In the early stages of research, Takano et al. established an IVIVC, with supersaturation characteristics in vitro dissolution of FTI-2600 and in vivo intraluminal concentration estimated from the pharmacokinetics data. The results provided clear evidence that not only the increase in the dissolution rate, but also the supersaturation phenomenon improved the solubility-limited absorption of FTI-2600 [145]. Another example reported that there is a good agreement obtained between the relative in vivo F and ABT-072 concentrations in octanol at 2 h. This indicates that the duration and extent of supersaturation of ABT-072 are directly related to the oral exposure in human subjects [100].

SDDS are difficult to achieve IVIVC at level A for these reasons: (1) due to precipitation, the in vitro dissolution curve of SDDS always decreases in the latter end and makes it hard to directly correlate with the continuously increasing in vivo absorption curve; (2) when adopting in vitro absorption compartment (octanol etc.), the absorption rate is usually limited by the area of contact surface, leading to deviation to the in vivo absorption rate; (3) the high variation of supersaturation leads to prediction error exceeding the allowance (15%). To overcome these disadvantages, biorelevant dissolution media and gastrointestinal tract biomimetic apparatus, such as those discussed above, are especially contributive to successfully established IVIVC [146]. In addition, some mathematical methods, such as Levy plots, may provide the proper scaling factor for the different absorption rates between in vitro and in vivo, according to our experience.

## 5. Discussion and Conclusions

Supersaturation is a promising strategy to improve oral bioavailability of poorly soluble drugs. In recent years, this study field has grown rapidly and become well established, especially in the pharmaceutical industry [147]. The rapid progression requires regular publication of summaries about its advances. The present article systematically introduces the main routes to generate supersaturation and the methods to inhibit precipitation, so that researchers are able to quickly understand the ways to employ supersaturation strategies in their study. Then, the development strategies and evaluation approaches to supersaturation were summarized covering in vitro, in vivo and in silico aspects. We believe that supersaturation strategy will be widely applied in oral formulations, and precipitation inhibitors should be included in pharmaceutics textbooks and catalogues at the same level as solubilizer and cosolvent. Besides oral formulations, supersaturation is possibly a strategy for enhancing other drug delivery routes, such as transdermal [148].

Contemporary researchers have preliminarily revealed the mechanism and elucidated the theory of supersaturation; however, there are still some problems to be solved. For example, is there a way to obtain the true degree of supersaturation of a pH sensitive drug in a condition with fluctuating pH? It is difficult to detect the equilibrium solubility along with the changing pH in the gastrointestinal condition, especially in the existence of polymers as the precipitation inhibitor. Then, is there a way to evaluate the re-dissolve of the precipitate? Following the absorption of the dissolved molecules, the re-dissolve rate of the precipitate obviously influences the bioavailability, but the precipitate may be quite different. In addition, there is a question of establishment of IVIVC to predict in vivo performance more efficiently, especially for level A IVIVC. Because in vitro simulation cannot always perfectly meet the complex in vivo variations, any small in vivo change could lead to a huge gap. We therefore hypothesize that the supersaturation/precipitation in the gastrointestinal tract is a Markov process. This means that the “future” and “past” of the process are independent of each other for a known “present”.

In the future, new supersaturation studies might develop from the following studies. More physiological data should be considered in in vitro studies to simulate the in vivo environment. In addition, understanding mathematical models at the molecular level by molecular simulation and particle growth limited in nano-scale may be the breakthrough point of the biopharmaceutical study on SDDS. Advancing material, instruments and technology will further complete the drug supersaturation theory. Integration with the theory and technology of self-assemble and supramolecule might extend supersaturation theory even further.

## Figures and Tables

**Figure 1 molecules-28-02212-f001:**
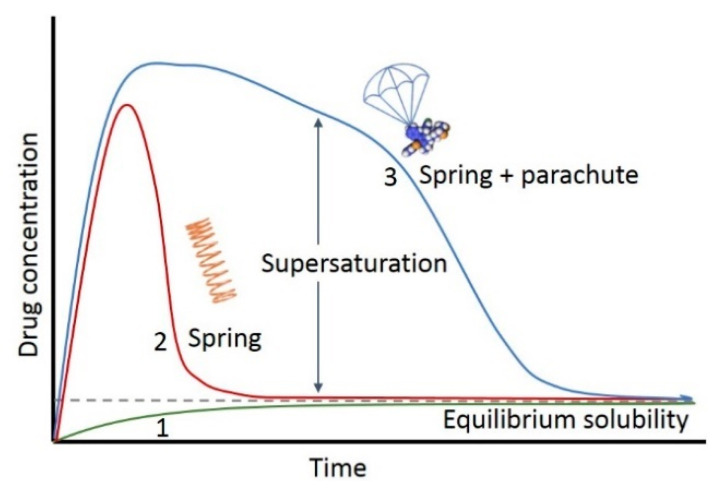
Schematic drug concentration–time profiles illustrating the supersaturation and precipitation process (“spring and parachute approach”) of a hypothetical drug. Profile 1: the dissolution of the most stable crystalline phase; Profile 2: the dissolution of a higher energy ‘‘spring’’ form of the drug in the absence of precipitation inhibitors; Profile 3: the dissolution of a higher energy “spring” form of the drug in the presence of precipitation inhibitors that act as a ‘‘parachute’’.

**Figure 2 molecules-28-02212-f002:**
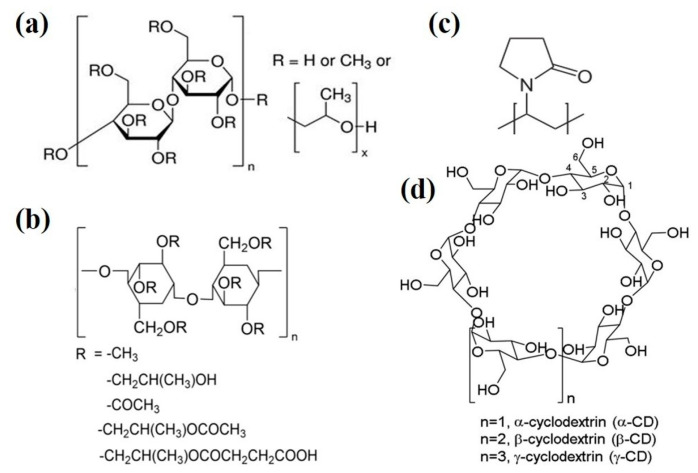
The molecular structures of commonly used precipitation inhibitors (**a**) Hydroxypropyl methylcellulose (HPMC), (**b**) Hydroxypropyl methylcellulose-acetate succinate (HPMCAS), (**c**) Polyvinyl pyrrolidone (PVP) and (**d**) Cyclodextrins.

**Figure 3 molecules-28-02212-f003:**
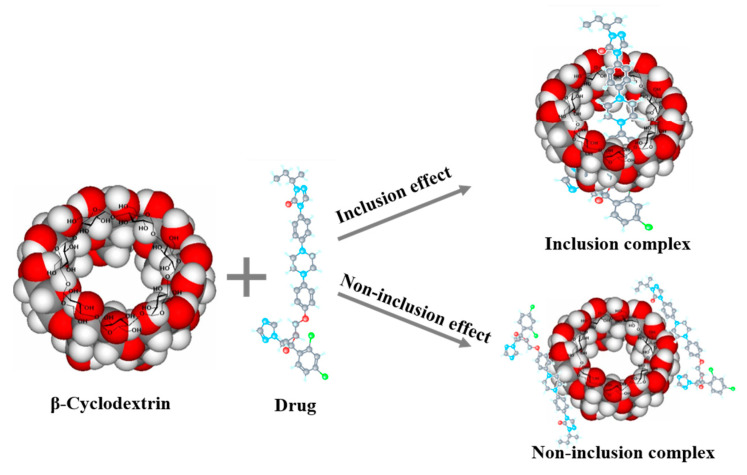
Illustration of inclusion complexes and non-inclusion of cyclodextrins.

**Figure 4 molecules-28-02212-f004:**
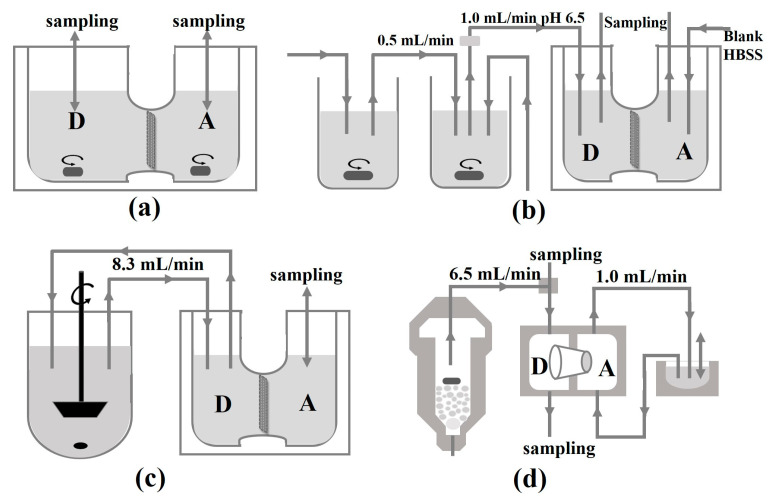
Schematic depiction of the published approaches for concomitant assessment of dissolution and permeation. (**a**) Dissolution/permeation system (D/P system); (**b**) the drug absorption prediction system; (**c**) continuous dissolution/Caco-2 system; (**d**) flow-through/penetration system. Donor compartments are denoted as “D”, acceptor compartments as “A”. Flow rates are provided where necessary.

**Figure 5 molecules-28-02212-f005:**
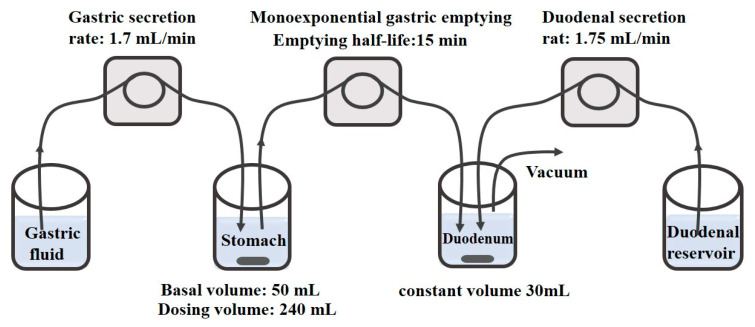
Schematic diagram of the simulated stomach duodenum model. Reprinted with permission from Ref. [6]. 2023, American Chemical Society.

**Figure 6 molecules-28-02212-f006:**
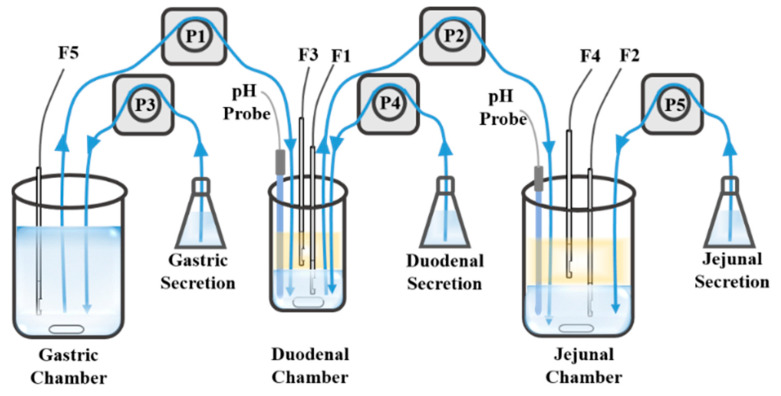
The schematic of BGIS. The blue blocks represent the aqueous phase, the yellow blocks represent the organic phase, F1–F5 represent the fiber optical probes, P1–P5 represent the pumps, the blue lines represent the tubes and the arrows represent the direction of the fluid. Adapted with permission from Ref. [115]. 2023, Elsevier.

**Figure 7 molecules-28-02212-f007:**
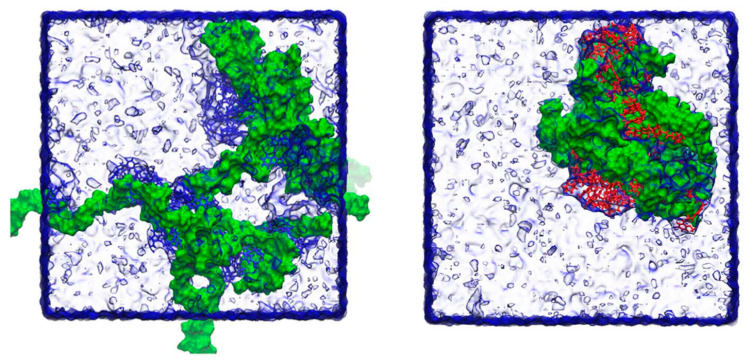
Snapshots of the final frame (at 100 ns) for the molecular dynamics simulations of indapamide (**left**) and glibenclamide (**right**) and HPMC at the low number of molecules. Indapamide molecules are indicated in blue, glibenclamide in red and HPMC in green, with surrounding water molecules rendered as a transparent surface. During the course of the simulations, individual drug molecules associate and disassociate from the polymer chains in both cases. Adapted with permission from Ref. [128]. 2023, Elsevier.

**Figure 8 molecules-28-02212-f008:**
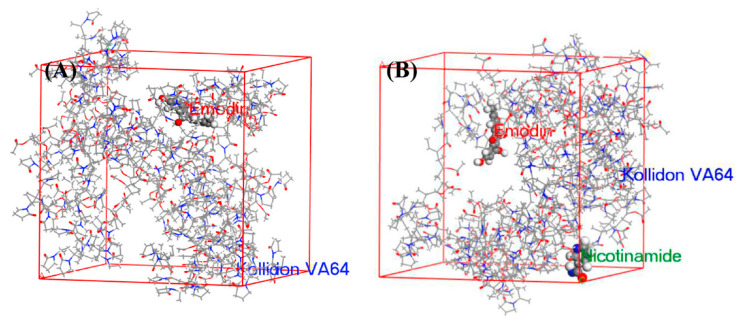
Snapshots of the final frame (at 100 ns) (**A**) Emodin and Kollidon^®^ VA 64 and (**B**) Emodin, Kollidon^®^ VA 64 and nicotinamide, of the molecular dynamics simulation. Adapted with permission from Ref. [132]. 2023, Elsevier.

**Table 1 molecules-28-02212-t001:** The pH values and the transit time at different anatomic sites of the human GI tract. Adapted with permission from Ref. [8]. 2023, John Wiley and Sons.

	Fasted State		Fed State	
Anatomic Site	pH	Transit Time (h)	pH	Transit Time (h)
Stomach	1–3.5	0.25	4.3–5.4	1
Duodenum	5–7	0.26	5.4	0.26
Jejunum	6–7	1.7	5.4–6	1.7
Ileum	6.6–7.4	1.3	6.6–7.4	1.3
Cecum	6.4	4.5	6.4	4.5
Colon	7.8 [9]	13.5	6.0 [9]	13.5

**Table 2 molecules-28-02212-t002:** Comparison of the enhancement effect among different SDDS.

		DS_max_	AUC Ratio	C_max_ Ratio	T_max_ Ratio	Permeability Ratio

ASDs		28.2	6.95	7.31	0.66	2.39
SEDDS		17.4	3.22	3.68	0.57	3.06
Mesoporous-based systems		47.4	4.52	4.63	0.80	/

Abbreviations: ASDs: Amorphous solid dispersion; DS_max_: Maximum degree of supersaturation; AUC: Area under curve; C_max_: Maximum concentration; T_max_: Time of Maximum concentration.

**Table 3 molecules-28-02212-t003:** The content and properties of several biorelevant media (mmol).

	FaSSIF	FeSSIF [81]	FaSSIF-V2	FeSSIF-V2 [82]	FaSSIF-V3 [83]	FaHIF [84]	FeHIF
Taurocholate	3	15	3	10	1.4	0.768	0.789
Taurodeoxycholate						0.381	0.283
Taurochenodeoxycholate						0.622	0.748
Glycocholate					1.4	2.766	2.585
Glycodeoxycholate						1.016	1.25
Glycochenodeoxycholate						2.281	2.371
Lecithin	0.75	3.75	0.2	2	0.035	2.06	5.81
Lysolecithin					0.315		
Cholesterol					0.2		
Sodium oleate				0.8	0.315		
Glycerol monoleate				5			
Buffer	Phosphate	Acetate	Maleate	Maleate			
pH	6.5	5.0	6.5	5.8	6.7	7.5	6.0
Osmolarity (mOsm/kg)	270	670	180	390	220	224	379
Buffer Capacity (mM/dpH)	12	76	10	25	5.6		

Abbreviations: FaSSIF (Fasted State Simulated Intestinal); FeSSIF (Fed State Simulated Intestinal); FaHIF (Fasted State Human Intestinal); FeHIF (Fed State Human Intestinal).

## Data Availability

Not applicable.

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
