# Peer review of "Supersaturation and Precipitation Applicated in Drug Delivery Systems: Development Strategies and Evaluation Approaches"

_molecules, 2023, doi:10.3390/molecules28052212_

Round 1
Reviewer 1 Report
The authors presented a paper entitled "Supersaturation and Precipitation Applied in Drug Delivery Systems: Development Strategies and Evaluation Approaches". Although the topic appears to be interesting, I believe there are no conditions for publishing the manuscript.
In particular, in the description of new drug delivery systems, e.g. section 2.2.1, too few studies have been presented. The co-amorphous formulations that have only been mentioned perhaps could have been better described.
In general, too few studies have been reviewed in each section to be considered exhaustive, and those few have been described marginally.
The period between lines 192-197 is not in the right place, reference is made to other routes of administration inherent in supersaturation formulations but this should be said at the end of the section or in a dedicated section.
Paragraph 3.2.1 is difficult to understand, you don't understand what you want to describe.
Too little is described about PVP and the rest of the inhibitors, and it would have been ideal to include some figures on the structure and/or mechanism in solution.
Section 4.1 seems to be unrelated to the rest of the work, the description of the media in relation to supersaturation formulations is important but too far into the merits of the techniques and there is little discussion of their influence on formulations.
The same in section 4.1.2, where the devices described are very common among those who study drug form release studies and therefore are not exclusive to those who work on the formulations of the review, and there is a tendency to go "off topic".
Reviewer 2 Report
The manuscript reviews the state of the art in the field of supersaturation drug delivery. The manuscript is well organized and structured, the level of English is good, and the content is useful to the reader. However, a point that is missing, is the authors' opinion. The authors should provide a discussion about the importance of the supersaturation research, possible future development, and also outline areas where more research has to be conducted.
Author Response
Response to Reviewer 2 Comments
Point 1: However, a point that is missing, is the authors' opinion. The authors should provide a discussion about the importance of the supersaturation research, possible future development, and also outline areas where more research has to be conducted.
Response 1: Thanks for your valuable suggestion. We have given opinions in some parts of this article, though they may be not conspicuous. We emphasized these opinions in the last section and discussed their importance, possible future development, and further research which has to be conducted.
Lines 669-703:
5.Discussion and Conclusion
【the importance】Supersaturation is a promising strategy to improve oral bioavailability of poorly soluble drugs. In recent years, this study field has grown rapidly and become well established, especially in the industry [151]. The rapid progression requires regular publication of summaries about its advances. The present article systematically introduces the main routes to generate supersaturation and the method to inhibit precipitation, so that researchers are able to quickly understand how to employ supersaturation strategies to their study. Then, the development strategies and evaluation approaches to supersaturation were summarized covering in vitro, in vivo and in silico aspects. We believe that supersaturation strategy will be widely applied in oral formulations, and precipitation inhibitors should be included in pharmaceutical textbooks and catalogues at the same level as solubilizers and cosolvents. Beside oral formulations, supersaturation is a possible strategy to enhance other drug delivery routes, such as transdermal [152].
【more research has to be conducted】The contemporary researches have preliminarily revealed the mechanism and elucidated the theory of supersaturation but there are still some problems to be solved. For example, how to get the true degree of supersaturation of a pH sensitive drug in a condition with fluctuating pH? It is difficult to detect the equilibrium solubility along with the changing pH in the gastrointestinal condition, especially in the existence of polymers as the precipitation inhibitor. Then, how to evaluate the re-dissolve of the precipitate? Following the absorption of the dissolved molecules, the re-dissolve rate of the precipitate obviously influences the bioavailability, but the precipitate maybe quite different. Beside, how to establish IVIVC to predict in vivo performance more efficiently, especially for level A IVIVC. Because in vitro simulation cannot always perfectly meet the complex in vivo variations, any small in vivo change could lead to a huge gap. We therefore hypothesize that the supersaturation/precipitation in the gastrointestinal tract is a Markov process. This means that the "future" and "past" of the process are independent of each other for a known "present".
【future development】In the future, supersaturation study might develop from the following studies. More physiological data should be considered in in vitro studies to simulate the in vivo environment. Besides, understanding mathematical models at the molecular level by molecular simulation and particles growth limited in nano-scale may be the breakthrough point of the biopharmaceutical study on SDDS. Advancing material, instruments and technology will further complete the drug supersaturation theory. Integration with the theory and technology of self-assemble and supramolecular technology, might extend the supersaturation theory even further.
【opinions】
Lines 283-287:
HPMCAS is not suitable as an active pharmaceutical ingredient (API) to be absorbed in the stomach or upper intestine (especially for weak basic drugs). Enteric soluble poly-mers Hydroxypropyl methylcellulose phthalate, cellulose acetate phthalate and Eu-dragit® have the same disadvantages.
Lines 488-489:
Furthermore, gastrointestinal organoids based on microfluidic chips are promising to be design and applied to study drug absorption and supersaturation.
Lines 548-550:
We propose that fiber optic detection system can be introduced into the intestinal content in situ detection like endoscope. Of course, fiber optical probe should be rede-signed as flexible and long enough to fit the access into gastrointestinal tract.
Lines 658-568 :
SDDS are difficult to achieve IVIVC at level A for these reasons: (1) due to precipitation, the in vitro dissolution curve of SDDS always decreases in the latter end, making it hard to directly correlate with the continuously increasing in vivo absorption curve; (2) when adopting in vitro absorption compartment (octanol etc.), the absorption rate is usually limited by the area of contact surface, leading to deviation to the in vivo absorption rate; (3) the high variation of supersaturation leads prediction error exceeding the allowance (15%).
Reviewer 3 Report
No: Molecules-2163394
Yanxiong Gan et al.: „Supersaturation and Precipitation Applicated in Drug Delivery Systems: Development Strategies and Evaluation Approaches”
The manuscript is a review article providing a good summary of the recent development in the subject indicated in the title. The summary is based on 128 cited references. Unfortunately in the present form the year of the publications is not shown, such way it is difficult to judge how many % of the references are form the last 5 years. The topic is undoubtedly current, since supersaturation and SDDS is a promising tool to overcome the poor absorption of BCSII drugs.
The content is well constructed, sufficiently concise and enough informative. It introduces the theory of supersaturation, how it can be generated and the methods for evaluations. In vitro, in vivo and in silico approaches and IVIVC are discussed.
I consider this review to be valuable but before being accepted for publication a thorough English correction is necessary by native speaker. The English is somewhere erroneous (e.g physiology conditions, biomimicking, etc) elsewhere obscure, obsolete, and at some places even it is not understandable.
Some critical remarks:
1. line 39-40. Give a more precise definition of supersaturation. What the authors mean on “normal circumstances”?
2. Table 1. the colon pH is about 8. Check the number in the cited paper.
3. Table 3. The buffer solution in FeSSIF is acetate not phosphate.
4. Figure 2. Give the meaning of a), b), c) d) letters in the capture.
5. line 415. “By using PAMPA separated side-by-side chamber” is nonsense. PAMPA is an assay based on 96 well-microplate, probably the artificial membrane can be applied not the PAMPA.
6. line 595-600. The text is confusing, has to be rewrite.
7. Ref. [61] and [66] are identical, remove the duplicate.
8. References are not in the journal format. They should be as:
Author 1, A.B.; Author 2, C.D. Title of the article. Abbreviated Journal Name Year, Volume, page range.
Author Response
Response to Reviewer 3 Comments
Point 1: line 39-40. Give a more precise definition of supersaturation. What the authors mean on “normal circumstances”?
Response 1:
Thanks for your constructive suggestions. We cited the definition of supersaturation from the classical article of Professor Patrick Augustijns, who are one of the leaders in the drug supersaturation study. Subsequently, the obscure words “normal circumstances” were avoid.
Lines 39-40:
Drugs in solution maybe at a concentration above their saturation solubility, that is, in a state of supersaturation [2].
Point 2: Table 1. the colon pH is about 8. Check the number in the cited paper.
Response 2:
We check the colon pH in the cited paper and found that the number is right as 6.8. Searching more papers, we found that the colon pH is indeed near to 8. The colon pH in the cited paper is in doubtful. So we adopt 7.8 from a paper published by professor Jennifer B. Dressman and professor Christos Reppas, who are the leaders in the biorelevant media study field. The result was emplyoyed to develop the commercial biorelevant media FaSSCoF, which is widely used in the dissolution study.
Point 3: Table 3. The buffer solution in FeSSIF is acetate not phosphate.
Response 3:
We are sorry to make this mistake, we corrected the buffer solution in FeSSIF to acetate.
Point 4: Figure 2. Give the meaning of a), b), c) d) letters in the capture.
Response 4:
We completed the meaning of a), b), c) d) letters in the capture.
Lines 449-451 :
(a) dissolution/permeation system (D/P system); (b) the drug absorption prediction system; (c) continuous dissolution/Caco-2 system; (d) flow-through/penetration system.
Point 5: line 415. “By using PAMPA separated side-by-side chamber” is nonsense. PAMPA is an assay based on 96 well-microplate, probably the artificial membrane can be applied not the PAMPA.
Response 5: Thank you for your suggestion and we improved the expression as “By using an artificial membrane separated side-by-side chambers”(in line 458).
Point 6: line 595-600. The text is confusing, has to be rewrite.
Response 6: We rewrite this part in lines 657-663:
SDDS are difficult to achieve IVIVC at level A for these reasons: (1) due to precipitation, the in vitro dissolution curve of SDDS always decreases in the after end, hard to directly correlate with the continuously increasing in vivo absorption curve; (2) when adopting in vitro absorption compartment (octanol etc.), the absorption rate is usually limited by the area of contact surface, leading to deviation to the in vivo absorption rate; (3) the high variation of supersaturation leads prediction error exceeding the allowance (15%).
Point 7: Ref. [61] and [66] are identical, remove the duplicate.
Response 7: We are sorry to make this mistake. The duplicate was removed.
Point 8: References are not in the journal format. They should be as:
Author 1, A.B.; Author 2, C.D. Title of the article. Abbreviated Journal Name Year, Volume, page range.
Response 8: We added Year of all the references by correcting the Endnote Style, e.g.:
- Aung, W. T.; Khine, H. E. E.; Chaotham, C.; Boonkanokwong, V., Production, physicochemical investigations, antioxidant effect, and cellular uptake in Caco-2 cells of the supersaturable astaxanthin self-microemulsifying tablets. Eur J Pharm Sci 2022, 176, 106263.
Point 9: I consider this review to be valuable but before being accepted for publication a thorough English correction is necessary by native speaker. The English is somewhere erroneous (e.g physiology conditions, biomimicking, etc) elsewhere obscure, obsolete, and at some places even it is not understandable.
Response 9: We are sorry for the imperfect English. The manuscript has been corrected and improved by native speaker Helen Van Oord, MSc, who lives in Melbourne, Australia, and experiences in academic English.
Reviewer 4 Report
In this review, the authors summarised recent development of supersaturation and precipitation applicated in drug delivery systems. Overall, this review article is well-structured, well-written and well-referenced, which will be of broad interests for readership of Molecules across a wide range of areas including pharmaceutical and medicinal chemistry, drug delivery, polymer science and so forth. Therefore, I would recommend its publication in Molecules after some minor edits.
1) Throughout this review, there is no schematic to show molecular-level mechanisms of drug supersaturation and inhibition of precipitation as well as molecule structures of several typical drugs. I would suggest the authors to add three or four figures to explain these mechanisms and chemistry behind this supersaturation strategy.
2) In recent years, supramolecular chemistry (host-guest chemistry in particular) also plays an important role in supersaturating drugs for specific therapy and treatment. Several key references are missing: e.g., Chem. Soc. Rev., 2017, 46, 7021; ACS Appl. Mater. Interfaces 2017, 9, 10, 8602; Chem. Soc. Rev., 2020,49, 2303.
Reviewer 5 Report
Thank you for a great opportunity to review the manuscript entitled "Supersaturation and Precipitation Applicated in Drug Delivery Systems: Development Strategies and Evaluation Approaches." I have some comments and suggestions for consideration to improve the manuscript quality.
Comment
This review article discussed strategies to develop supersaturating drug delivery system (SDDS) and approaches to evaluate in vitro and in vivo performance of SDDS formulations. I believe that this review help researchers understand and make deep discussion on SDDS approaches to improve bioavailability of poorly water-soluble drugs classified as BCS class II.
Minor suggestion
1. If possible, I would like to recommend authors to describe liquid-liquid phase separatoin that has been studied in the field of supersaturation. For example, in the section of amorphous solid dispersion.
2. In the section of pharmacokinetic simulation, it may be helpful for readers deep understanding to describe the advantage/disadvantage of each simulation approach more clearly.
Author Response
Response to Reviewer 5 Comments
Point 1: If possible, I would like to recommend authors to describe liquid-liquid phase separatoin that has been studied in the field of supersaturation. For example, in the section of amorphous solid dispersion.
Response 1: Thanks for your sugestions that we learned more about liquid-liquid phase separation in the supersaturation. We describe liquid-liquid phase separation in the section of amorphous solid dispersion. In addition, we also take liquid-liquid phase separation as an example in section 3.2.2 HPMCAS.
Lines 139-157:
Ashwathy et al. recently summarized the mechanism of phase separation about supersaturation [16]. Furthermore, liquid-liquid phase separation (LLPS) has been found to be vital for supersaturation maintenance of ASD [17]. As the free drug concentration con-sistently rises, the original homogeneous system is transformed into a heterogeneous system composed of a supersaturated solution phase and a drug aggregation super-cooled liquid phase. This process produces highly concentrated (nano) droplets/particles, which cannot be absorbed directly but may act as drug reservoirs shuttles, or both to carry drug molecules effectively in the mucus layer [18, 19]. A study prepared paclitaxel/polymer amorphous solid dispersion by solvent casting, to evaluate the effect of supersaturation on the oral absorption of paclitaxel. The dissolution rate was remarkably improved, reaching a DS about 100 and maintaining it for more than 2 h, with a 1.78-fold increase in relative oral bioavailability [20]. Kyosuke et al[18] studied the relevance of LLPS of supersaturated solution in oral absorption of albendazole from ASD. In the absence of the polymer, LLPS of Albendazole occurred at 1.4 µg/mL, which was more than 10-fold the solubility, whereas it increased to 7.2, 7.0, and 3.8 µg/mL with PVPVA, HPMCAS, and Eudragit, respectively. The LLPS concentration is well correlate with AUC in rats (R2 = 0.9421). A better control over both the solid-state stability and super-saturation generation and maintenance will lead us to delivery systems having desirable and predictable properties and a practical option to solubilize the “difficult to solubilize” drugs [21].
Lines 276-278:
Recently, Masafumi et al. [54] confirmed that HPMCAS is able to inhibit liquid–liquid phase separation in the supersaturation process of naftopidil amorphous solid dispersion at 41µg/mL.
Point 2: In the section of pharmacokinetic simulation, it may be helpful for readers deep understanding to describe the advantage/disadvantage of each simulation approach more clearly.
Response 2: Thanks for your interest in pharmacokinetic simulation related to supersaturation and providing constructive suggestions on this part. We rewrote this part and empased on the advantage/disadvantage of each simulation approach.
Lines 599-564:
GastroPlus™, is developed based on advanced compartmental absorption and transit (ACAT) model, which is composed with 9 chambers of gastrointestinal tract, considering 6 state of drug in the dissolution and absorption. Taking this advantage on the meticulous physiological condition, it provides single or multiple first order exponential precipitation or a complete mechanistic nucleation and growth model that can account for formulation effects due to nucleation inhibitors and solubilizers [133].
Lines 609-616:
However, there are some disadvantages: it ignored some special forms of drugs, such as the salt and stereoisomers; so many drug parameters are required to establish the model; the accuracy of the model is highly dependant on the modeling strategies [135].
Simcyp® is based on advanced dissolution absorption and metabolism (ADAM) model, which is composed with 7 chambers of gastrointestinal tract. It takes advantages in absorption kinetics with respect to disintegration time, DS and precipitation rate.
Lines 618-620:
Its disadvantages were reported as: default models and parameters were allowed, potentially leading to wrong conclusions; unable to distinguish between the drug salt and the base present in different proportions, leading to more probable failure in BE testing [137].
Lines 622-623:
GI-Sim is also based on ADAM model. Its advantage includes giving consideration to particle growth, amorphous or crystalline precipitation and supersaturation parameter.
Lines 630-631:
So, its disadvantage is also similar to that of Simcyp®.
Round 2
Reviewer 1 Report
Authors addressed all the proposed suggestions.